# Conjugated Copolymers through Electrospinning Synthetic Strategies and Their Versatile Applications in Sensing Environmental Toxicants, pH, Temperature, and Humidity

**DOI:** 10.3390/polym12030587

**Published:** 2020-03-05

**Authors:** Loganathan Veeramuthu, Manikandan Venkatesan, Fang-Cheng Liang, Jean-Sebastien Benas, Chia-Jung Cho, Chin-Wen Chen, Ye Zhou, Rong-Ho Lee, Chi-Ching Kuo

**Affiliations:** 1Institute of Organic and Polymeric Materials, Research and Development Center of Smart Textile Technology, National Taipei University of Technology, Taipei 10608, Taiwan; anloga947715@gmail.com (L.V.); manikandanchemist1093@gmail.com (M.V.); frank62112003@yahoo.com.tw (F.-C.L.); Benas.jeansebastien@gmail.com (J.-S.B.); cwchen@ntut.edu.tw (C.-W.C.); 2Institute for Advanced Study, Shenzhen University, Shenzhen 518060, China; yezhou@szu.edu.cn; 3Department of Chemical Engineering, National Chung Hsing University, Taichung 402, Taiwan; rhl@dragon.nchu.edu.tw

**Keywords:** conjugated copolymers, electrospinning, nanofibers, sensors

## Abstract

Conjugated copolymers (CCPs) are a class of polymers with excellent optical luminescent and electrical conducting properties because of their extensive π conjugation. CCPs have several advantages such as facile synthesis, structural tailorability, processability, and ease of device fabrication by compatible solvents. Electrospinning (ES) is a versatile technique that produces continuous high throughput nanofibers or microfibers and its appropriate synchronization with CCPs can aid in harvesting an ideal sensory nanofiber. The ES-based nanofibrous membrane enables sensors to accomplish ultrahigh sensitivity and response time with the aid of a greater surface-to-volume ratio. This review covers the crucial aspects of designing highly responsive optical sensors that includes synthetic strategies, sensor fabrication, mechanistic aspects, sensing modes, and recent sensing trends in monitoring environmental toxicants, pH, temperature, and humidity. In particular, considerable attention is being paid on classifying the ES-based optical sensor fabrication to overcome remaining challenges such as sensitivity, selectivity, dye leaching, instability, and reversibility.

## 1. Introduction

The major goal of this review is to highlight the importance of conjugated copolymeric systems and its various types of sensory applications. The sensitivity and selectivity of the sensors are influenced by synthetic and fabrication techniques with respect to methodology and ratios of polymeric blends. A conjugated polymer is a macromolecule with alternated π bonds and sigma bonds, which finds fascinating optical and optoelectronic applications. Several conjugated aromatics such as spiropyran and diarylethene groups functioning as organic photochromic devices due to the bond forming and cleaving ability on exposure to light [1,2]. Carbazole is another class of conjugated polymer witnessing the good electrical and fluorescent behavior because of its extensive π conjugation [3]. Unlike conventional polymers, conjugated polymers possess better electron mobility because of the delocalized Pz orbitals. Aromatic conjugated polymers obeyed Huckel’s rule with exceptional stability. The π bond of Pz orbitals in the benzene ring overlaps with neighboring Pz orbitals lying perpendicular to each other favoring the delocalization of π electrons [4].

The last few decades have seen a major thrust in the development of conjugated polymer based organic thin film transistors, organic photovoltaics, battery electrolytes, organic light-emitting diodes (OLEDs), and sensors [5,6,7,8,9,10]. Organic photovoltaics utilize organic conjugated polymers due to their high absorption coefficient and band gap tuning by modifying the chemical moieties to cover the entire ultraviolet-visible (UV-Vis) absorption spectrum [11]. The removal of environmental toxicants or effluents from the water, air, and, soil is urgently required. These toxicants are expelled by a variety of technologies, such as adsorption, filtration, ion-exchange membranes, coagulation, precipitation, sedimentation, and other biological enzymatic approaches [12,13,14,15,16]. Before and after the removal of toxicants, it is mandatory to measure contamination levels, which enables the evaluation of removal technique efficiency. Sensors are described as devices that can respond to analytes or external environments (metal, pH, temperature, humidity, etc.) with good sensitivity and selectivity without any considerable interference or contamination of the analyte. The sensors output can be monitored with optical responses, such as fluorescent off–on, on–off, and color changes, or by other coupled electrical read out devices [17,18,19].

Although conjugated polymers (CPs) have received attention in the research community, we herein review the conjugated copolymer (CCP)-based sensory applications due to its high demand in assessing environmental safety and monitoring sectors. Among various applications, sensory application is dominating due to the emerging organic chemist skills on tuning the CPs functionalities and fine-tuning it to the desired applications. Various sensing mechanism and modes of sensing furnish unprecedented CCPs sensory responses with ultrahigh sensitivity and selectivity [20,21,22,23,24,25,26]. Huge no of analytes have been monitored with good sensitivity utilizing the high end instrumental techniques such as high performance liquid chromatography (HPLC), gas liquid chromatography (GLC), gel permeation chromatography (GPC), energy dispersive X-ray spectroscopy (EDS), inductive couple plasma-atomic emission spectroscopy (ICP-AES), UV-Vis, and so on. It is always desirable to use less expensive, user friendly, lightweight, reusable, portable devices to meet low end user utilization. By the above perspective, fluorescent and colorimetric conjugated polymeric strips can function as a promising candidate to meet all user demands.

We herein review the challenging aspects of designing highly responsive optical sensors through facile synthetic strategies, sensor fabrication, mechanistic aspects, and sensing modes. Moreover, we review recent sensing trends in monitoring environmental toxicants, pH, temperature, and humidity. Classification of electrospinning (ES)-based optical sensor fabrication and its potential in overcoming the existing sensory challenges such as sensitivity, selectivity, dye leaching, instability, and reversibility are also discussed. CCPs’ sensory applications and their potential future applications are also presented along with the recent developments.

## 2. Need of Electrospinning Technique

In early stages, considerable attention was paid on the solution based optical sensors and good level of detection and sensitivity ranges was achieved. The solution-based optical sensors lag behind in sensory application because of the solvent restrictions, pH conditions, and contamination imposed by the sensory solution to the analyte [27,28]. As a consequence, solid optical sensor systems are highly appreciated in terms of flexibility, easy analysis without contaminating the analyte, reproducibility and portability [29,30,31]. Thin films can be formed by co called techniques like drop casting, spin casting, dip coating, template assisted, and printing methods. Polymeric films can be formed on the anode surface by electrochemical oxidative polymerization. The quality of the so called polymeric films relies on the monomer structure, electrode material, solvents and temperature factors [32,33]. The sensory thin films are reliable and the formation is stable enough to achieve the reproducible results. Still, the limit of detection (LOD), response time, reproducibility, sensitivity, and selectivity remains limited, and it does not meet the state-of-the-art values because of lowered surface to volume ratios and lack of roughness. In the viewpoint of fluorescent optical properties, thin films were not good enough; they possess numerous surface defects that quench the photoluminescence quantum yield (PLQY) and radiative rates [34,35]. These limitations have been overcome by utilizing the electrospinning technique which can perish all the ill effects prevailed on thin films [35,36].

## 3. Electrospinning Technique

Electrospinning (ES) is a simple process to produce the continuous nanofibers by applying an electric voltage through the metallic syringe (Scheme 1). The applied electric field facilitates the formation of superfine nanofibers by overcoming the polymeric viscosities and getting deposited onto the collector substrate [37,38,39]. The conventional methods like drawing, template synthesis, phase separation, and self-assembly cannot produce continuous nanofibers with good morphological features. Numerous limiting factors with the abovementioned methods limited the production of nanofibers. ES technique finds many applications due to its synergistic control over the formation of continuous nanofibers with nanometer sized diameters with reduced hysteresis [40,41].

Various factors influence the formation of bead free smooth nanofibrous structures such as spinning condition, polymer solution, and environmental conditions. The spinning condition includes the distance between the needle and collector, flow rate, needle diameter, and applied voltage [42,43,44,45]. The distance between needle and the collector crucially alters the flight time governing the solvent evaporation with considerable effects on the nanofibers (NFs) diameter [46]. The longer the distance facilitates the formation of thinner NFs. In contrast, Kim et al. observed the opposite trend, and it is ascribed to the restriction with elongation due to the increased viscoelastic forces [47]. Interestingly, the distance factors do not affect the fiber dimension where the water solvent employed cannot escape within a shorter period of spun time [48]. In short, the defect-free NFs can be obtained by modifying the distance between the needle and the collector indirectly influencing the flight time [49].

The polymeric solution factors, such as concentration, molecular weight, solvent, conductivity, viscosity, and surface tension, will affect the morphological features. In general, many CPs furnish rigid backbones with higher crystallinity that consequently exhibit the poor solubility with common solvents hindering its ES processability. The doping works better in overcoming solubility issues, for example, poly (3,4-ethylenedioxythiophene) polystyrene sulfonate (PEDOT PSS) forms conductive continuous pathways upon addition of dimethyl sulphoxide (DMSO) and isopropyl alcohol (IPA) solvents [50]. Co-solvent utilization improved the highly ordered homogenous fibers with a smoother morphology. Addition of low vapor pressure co-solvent effectively retards the thermodynamic driving force for liquid–liquid phase separation that modifies the solidification of the nanofibrous membranes (NFMs) [51]. Furthermore, temperature and relative humidity % can have an adverse effect on the spun nanofibers. The effect of all the as-described factors on the fiber morphology were considered and discussed elaborately in the previously published literatures [52,53,54,55].

## 4. Electrospun Nanofibers in Sensing Applications

The removal of environmental toxicants has been made possible by numerous techniques such as distillation, floatation, hand picking, enzymatic degradation, ion exchange, coagulation, flocculation, precipitation, electrochemical methods, and so on. Electrospun nanofibers (ESNFs) have high demand in producing air and water filters, and they are being commercialized due to their production efficiency in creating diameter downsized continuous nanofibers [55]. Additionally, it offers good control in creating diverse morphological structures (including core-shell, hollow, porous, and nano nets), which are less time-consuming, easy to handle, and portable. These ESNFs also generate no waste and are easy to collect in 3D structures [37,55,56]. Such features have made adsorption enabled toxicant removal through ESNF a highly valuable due to their higher surface to volume ratios [12,57,58].

Apart from removal, sensing is much concerned in terms of environmental safety hygiene and health management. A sensor is highly valued in the aspects of good range of sensing, lower LOD, higher sensitivity, selectivity, lower response time, reversibility, and stability in the sensing medium. The ESNFs are susceptible to various surface functionalization, immobilization, and other post-treatments in fine tuning its sensory applications [59,60,61]. Two different major strategies were employed to improve the sensing characteristics such as creating ESNFs with porous structures and ESNFs with sub-micro and nanostructures, which can influence the surface to volume ratio considerably [57,61]. For creating porous structures, solvent choice is the major contributor in designating the device sensitivity, selectivity, and response time. Solvent influences the spinning process in three ways: (i) solvent compatibility with the dissolved polymer, (ii) solvent viscosity after blending the polymer, and (iii) solvent volatility [62,63]. Nanoporous NFMs achieved using low boiling solvents cause spontaneous temperature difference on spinning and water vapor adsorption onto the polymer jet surface. Adsorbed water vapors and residual solvents left the jet surface to harvest the polymer rich phase matrix with nanoporous architecture [64]. Good solvent and bad solvent combinatorial approaches worked under the basis of volatility and water miscibility producing surface and internal porosity depending on the choice of solvents [65].

The strong stretching forces that accompany the ES process create a polymer chain orientation along the fiber axis. The excellent orientation and microscopic alignment factors observed in the ES process are not present in thin film fabrication processes such as drop casting and spin casting because of their different working mechanism. Such contributions provide the ESNFs with interesting anisotropic optical properties. CCPs have acquired greater interest attributed to the delocalized π electrons that reflects with its optical and electrical properties abruptly. These optoelectronic properties of CCPs can be harvested by utilizing synthetic procedures such as oxidative polymerization, free radical polymerization, electrochemical polymerization, and coupling reactions. Constructing the CPs with desirable optical and electrical characteristics is made viable through their synthetic flexibilities. CPs finds difficulty in generating ESNFs due to its poor miscibility so that CPs were framed as CCPs by grafting, adding functionalities onto the monomers and preparing random or block copolymers to employ them as sensors. Furthermore, multisensory applications are made possible with CCPs without deteriorating the stability and responsiveness towards analytes. Such outstanding research developments and breakthroughs were compiled with the utmost care to elucidate the sensing mechanisms, mode of sensing, classification of sensory systems, and their promising potential in sensing the analytes.

## 5. Sensing Mechanism and Mode of Sensing

Optical sensors in general follow various sensing mechanisms such as conformational change, chain aggregation, Forster or fluorescence resonance energy transfer (FRET), inner filter effect (IFE), metal complex formation, or intermolecular charge transfer (ICT) to produce the various naked-eye optical responses, such as off–on, on–off, and colorimetric changes (Scheme 2).

The conformational change in the polymer backbone occurs essentially because of the solvent system, doping levels, and surface modifications. Chandra et al. used polyaniline (PANI)-coated fiber optics with human immunoglobulin immobilization (assisted by glutaraldehyde cross-linker) in fabricating highly sensitive immunosensors that contrast with traditional protonation and deprotonation mechanisms [22]. Side-chain engineering in conjugated matrices is highly promising in generating colorimetric sensors because of its physical property variation according to the molecular conformation and packing. Chen et al. devised a Fe^3+^ sensor that functions in a 100% aqueous medium composed of anionic water-soluble poly(3,4-propylenedioxythiophene) derivative with modified carboxylic side chains. This work conveys the importance of conformation changes in producing the colorimetric response (purple to faint yellow) in response to Fe^3+^ ions [66]. The number of conjugated repeated units greatly influenced the sensitivity, and the small molecules require higher concentration of salt to induce the twisted complex side reaction equilibrium shift [67]. The conformational changes have been effectively studied with the aid of UV–visible, proton nuclear magnetic resonance (H^1^ NMR), and Fourier transorm infrared spectroscopy (FTIR) characterizations.

Aggregation caused quenching emission is commonly observed in many of the fluorescent organic molecules and in many recent perovskite-based devices. In 2001, Tang et al. reported a phenomenon called aggregation induced emission (AIE) with a propeller-shaped nonemissive organic molecule on dissolving with solvents forming emissive aggregates. The proposed mechanism suggests the curtailment of molecular rotations made possible with aggregate formation leading to reduction of non-radiative pathways [20,21]. Nanoaggregates were successfully employed in detecting nitroaromatics because of the selective aggregative responses provided by the conjugated compound [68].

Forster or fluorescence resonance energy transfer (FRET) is another interesting mechanism that was introduced more than half a century ago. It is an electrodynamic process that occurs between the excited donor fluorophore and the ground state acceptor. The FRET concept works very well in sensing the analytes because of its simplicity. The excellent overlap between the fluorescent emission (chromophore) and absorption spectrum (analyte), steady-state photoluminescence (PL), and time-resolved PL measurements affirms the existence of the FRET system, and its quenching efficiency can be altered with the concentration of analytes [23,69,70].

Inner filter effect (IFE) is closer to FRET in concept. We can differentiate IFE as a static quenching type, whereas FRET belongs to the dynamic quenching type. IFE works if there is a spectral overlap between chromophore emission or excitation and analyte absorption. Tanwar et al. studied the type of quenching with time resolved PL (TRPL) by monitoring the decay constants through which the solvent system concludes the type of sensing mechanism followed [25].

In particular for metal ion sensing, complex formation mechanism is ideal and it is followed for detecting several metal ions specifically with good degree of sensitivity. Especially in conjugated molecules, the electronic clouds delocalized with the chains can be altered using various functional groups enabling the sensing of various targeted metal ions [71,72,73].

Intermolecular charge transfer (ICT) is based on electron donors and acceptors, and it governs various classes of technologies such as organic light emitting diodes (OLEDs) and solar cells. On interaction with donor or acceptor groups of organic conjugated molecules, depending on the charge distribution and conjugation extent, the emission intensity might increase or decrease or even lead to highly variable color changes [26].

Sensing mechanism considerably influences the sensory response, and this is termed as the mode of sensing. Mode of sensing in optical sensors can be broadly classified as (i) fluorescent off–on type, (ii) fluorescent on–off type, and (iii) colorimetric type. The first two types are within the same emissive wavelength but the intensity decreases or increases, whereas the colorimetric type can exhibit the change in emission wavelength that can create contrasting colors in response to the sensing analyte.

## 6. Classification Based on Sensory Systems

Figure 1 illustrates the synthetic strategy and fabrication of the sensory ES nanofibers and herein we classify the optical sensors as follows.
Physical blended ES-based sensory nanofibersSurface functionalization- and grafting-based sensory nanofibersReversible addition fragmentation chain transfer (RAFT) and atom transfer radical polymerization-based ES sensory nanofibersRandom copolymer-based ES sensory nanofibersBlock copolymer-based ES sensory nanofibers

## 7. Physical Blended Electrospinning (ES)-Based Sensory Nanofibers

Physical blending is one of the most widely adopted approaches for generating optical sensory probes because of its ease of processing, compatibility, low cost, and nontoxic features, all of which enable physically blended ES sensory nanofibers as an acceptable strategy. This method remains reliable, and its evolution in sensing various analytes is discussed with the aim of improving its sensitivity and selectivity.

Lee et al. [74] fabricated a conjugated polydiacetylene (PDA)-embedded electrospun fiber mat to differentiate adulterated gasoline from pure gasoline. Diacetylene (DA) monomers were randomly distributed in a viscous organic solvent before electrospinning. As the solvent evaporated during fiber formation, the self-assembly of DA monomers occurred (Figure 2a). On exposure to adulterated gasoline, which contains toluene, the protective polystyrene (PS) polymer matrix disintegrated and PDA molecules exhibited solvatochromic behavior and colorimetric responses were observed. In case of methanol adulterated gasoline, the polyacrylic acid (PAA) polymer matrix underwent dissolution. This process relies entirely on the solubility parameters, polarity, and strong molecular interactions.

Recently, PDA embedded in ES polyvinylidene fluoride (PVDF) NFs has played a crucial role in generating dual colorimetric and piezoelectric responses (Figure 2b). PDA-incorporated NFs tend to possess a larger diameter and promote polar beta-phase growth. PVDF exhibits five crystalline conformations, among which trans conformation is highly demanded because of its piezoelectric properties. Achieving such promising conformations is made possible by facile ES technique with PDA; it additionally supports the optical sensory responses. The increment in beta phase is on the account of the polar carboxyl group of 10,12-pentacosadiynoic acid (PCDA), and the conformational presence can be analyzed through FTIR and wide-angle X-ray diffraction (WAXD). Under annealing conditions, the PCDA-conjugated backbone might disrupt the aligned molecular structures, ultimately leading to inferior beta phase. Blue phase enriched PDA ES nanofiber displayed greater sensitivity than red phase PDA because of the presence of higher beta-phase fractions [75].

Conjugated polymers are not easy to spin into ES fibers because of their rigid macromolecular structures and lack of side-chain functionalities. The tight coiled conformations aroused several attempts in way to solve the brittleness and mechanical instability issue which is common in pristine CPs. Using insulating polymeric host mechanical stability was improved considerably without compromising the sensing factors [77]. The synergistic role of polydiarylfluorenes side chain length and ES stretching effect deliberately facilitates the polydiarylfluorene β-phase formation demonstrating the elevated sensitivity [78]. Poly((9,9-bis(3′-(*N*,*N*-dimethylamino)propyl)-2,7-fluorene)-alt-2,7-(9,9-dioctyl-fluorene))-containing tertiary amine group physically interacts with carboxylated rubbers manifests the fluorescent elastomeric fibers with good stability and mechanical strength [79]. Fluorene-containing poly(aryl ether nitrile) synthesized using aromatic nucleophilic substitution polymerization exhibits good solubility because of the existing bulky pendant groups. Such structural modifications positively contributed to electrospinning process, producing better thermal stability and water repellency [80]. Fan et al. [81] reported on a receptor-free sensory device with excellent adsorbability based on conjugated polymers. Formerly designed sulfonium salt polymer precursor (pre-PPV) [82] can be easily blended on to various solvents and displays good compatibility with polyimide and polymethyl methacrylate (PMMA) polymers to capitalize on the favorable sensitivity and selectivity. Not having a receptor provides space to adsorb the analyte conformally and thereby affects the emission spectra. Compared with Poly(phenylenevinylene)(PPV)/P1, PPV/PMMA is highly sensitive and produces fluorescent off–on (>4 times enhancement) in response to 20 nM Cu^2+^. In addition, on–off fluorescent quenching occurs with 20 nM Fe^3+^ solutions. The higher fluorescent responses by the PPV/PMMA sensory probe are attributable to the porous secondary network structures. The merged fibrous morphology is possibly due to the thermal treatment, which is close to the glass transition temperature Tg of PMMA matrix.

Organic solvents pose a threat to industry employees and it is a crucial factor for health and safety regulations. Their leakage, improper recycling, and disposal have considerable effects on human health, potentially causing brain disorders, liver damage, kidney damage, etc. Conjugated PPV moiety can change the π–π intermolecular interactions that might aid in the reduction of the fluorescent self-quenching factor. The water-soluble PPV precursor polymer and polyvinyl alcohol (PVA) were electrospun and cross-linked with glutaraldehyde at room temperature to form PPV/ crosslinked PVA (CPVA) sensory membranes. The strong fluorescent on–off was observed due to the π–π interaction between the aromatic solvent and PPV, which enables it to easily distinguish aromatic organic vapors from other aliphatic solvents. The non-cross-linked PPV/PVA and cross-linked PPV/CPVA membranes’ sensory responsive behavior varies considerably in aqueous medium because of the unstable nature of PPV/PVA matrices. The cross-linked structures improved the reversible characteristics and they are superior to thermal treated nanostructures evidenced by their improved hydrophobic characters [83].

Food adulterations can cause severe damage to human health and lead to genetic disorders. Various dyes or colorants have contributed substantially in terms of production and good stability. The major concern in dye selection is their biodegradability. Sudan dyes are phenyl azo functional dyes used as additives in several cosmetics, plastics, paints, garments, and even in some food ingredients. The critical issue with sudan dye is its water-insoluble nature that possibly causes complications in the digestion and excretory system. The pre-PPV is blended with PVA and it is cast into films and NFs using drop casting and electrospinning respectively which is further cross-linked with glutaraldehyde. Pre-PPV mass percentage varied in the PPV/CPVA film, and membrane and morphological factors variations were compared and optimized. The well-matched absorbance of Sudan dye with PPV/CPVA membrane excitation and emission spectra can give rise to the IFE mechanism, thus resulting in a different extent of fluorescent quenching. The optimized PPV/CPVA membrane exhibits an appreciable LOD of 73.39 ng/mL with 10 cycles of reversibility [84].

Childhood cirrhosis, Menkes syndrome, prion disease, Parkinson’s disease, and Wilson disease are characterized by abnormal Cu^2+^ levels [19]. Ding et al. sensed Cu^2+^ with an ultralow LOD of 1 ppb and produced visible color change from white to blue without any interference from common metal ions. Cu^2+^ sensing is essential because Cu^2+^ is toxic to humans and it causes suffocations in the gastrointestinal and kidney systems. PANI CP has unique characteristics, such as its extent of protonation and oxidation (i.e., leucoemeraldine base (fully reduced), emeraldine base (half oxidized), and pernigraniline base (fully oxidized)), that can be used for sensing toxicants. PANI emeraldine base (EB) was chemically polymerized using aniline monomer with ammonium persulfate (APS) oxidant in acidic medium. PANI EB was blended with polyamide-6 (PA-6) to form ES nanofibrous membranes, and then, it was reduced to PANI leucoemeraldine base (LB) (fully reduced) with hydrazine reductant (Figure 2c). The selectivity in sensing Cu^2+^ is ascribed to its oxidation, and reduction potential matching and the presence of Cu metal–PANI complexes were reported. Furthermore, pH-dependent reflectance studies have revealed the strong protonation doping process in PANI [76].

Liquefied petroleum gas (LPG) leakages lead to severe accidents and cause harm to lives. Due to the emergence of LPG in automobiles, it is essential to sense for LPG leakage. This is both a personal and public safety concern. Traditionally prepared PANI polymer converted into emeraldine base (PANI EB) using ammonium hydroxide, and it is further protonated with D-camphor-10-sulphonic acid (CSA) to improve the conductivity and solubility for electrospinning. Hybridizing the PANI with ZnO is made through PANI EB blending with sol–gel ZnO along with 10% polyethylene oxide (PEO) solution [85]. More research concerns have to be paid on reducing the response time and improving the reversible cycles to reach the real-time applications. Therefore, the design of LPG optical sensors with better response and reversibility represents a great research opportunity.

## 8. Surface Functionalization and Grafting Based Sensory Nanofibers

Despite employing blending strategies, surface functionalization of CPs is promising and it enables easy binding thereby improving the sensitivity and selectivity. Either functionalizing the spun carrier matrix or functionalizing the dye conquers the sensing device with good stability and ultrafast responsive characters. The surface modification strategy is to endow specific functionalities with well-defined characteristics to the original polymer with desired characteristics. Surface modification of ES NFs after electrospinning is desirable as it affords a greater extent of functionalities that are exposed on the surface. From such highly exposed nanostructures, it is appropriate to expect ultra-high response possibly due to facile accessibility experienced by the analytes [86].

Poly(hydroxyethyl methacrylate) (pHEMA) hydrogel formed by radical polymerization and formed cross-linked interpenetrating chitosan networks. Furthermore, it was subjected to surface functionalization using spiropyran with –COOH functionality to establish amide linkage with chitosan interpenetrated networks. The optical responses arouse due to the ring opened merocyanine state (upon UV exposure) and protonated merocyanine state (on acidic conditions) [87]. Commercial PU fibers were polymerized in situ to produce compact polypyrrole layers with good conformability to be employed to sense chloroform vapors. Adsorption was duration altered to find the optimum compact polypyrrole surface. The resistive sensor worked on the basis of polyurethane (PU) polymer swelling behavior with chloroform vapors leading to higher resistance. The resistance change was rapid (36 s) and observed LOD was 150 ppm [88]. Recently, terpyridine (tpy) derivatives received much attention due to its binding efficiency and better positioned ring nitrogen atoms. Tpy functionalization on conjugated polymers affects the optical sensory characteristics and it is compared with and without tpy functionalization [19].

As discussed in our introduction, stability of the sensing chromophore is mandatory, and it should not disintegrate into the detection medium while sensing the analyte. This instability might severely affect the stability and reversibility of the fabricated solid-state sensors. Recently, Li et al. [3] fabricated amino functionalized glass and electrospun freshly synthesized 5-(N-carbazole styryl)-1,3-dimethyl-barbituricacid (CB) along with polymeric carrier polystyrene (PS) shortly termed (CB-PS)/(G-NH_2_) sensors to detect picric acid in aqueous medium (Figure 3a). The amine functionalized glass-based sensors performed better than conventional glass-based sensors with an ultralow LOD of 153 ppb. The high selectivity towards picric acid is due to the larger driving force differences as it follows the excited electron transfer from fluorophore lowest unoccupied molecular orbital (LUMO) to the analyte picric acid. This results in a 69% fluorescent turn off. Apart from stability and sensitivity, the dynamic response of the sensor also governs its supremacy in finding its real-time applicability.

The electrostatic interaction between amine functionalized ES PVA-silica NFs produces good mechanical stability. Highly emissive quinoxaline-based CP dots, immobilized onto the PVA–silica NFs, produced rapid turn off fluorescence in response to the presence of chemical warfare reagent. The dots on the unique fiber nanostructure formed by taking advantage of electrostatic interaction between charged NFs and CP dots due to the successful amine functionalization achieved by 3-minopropyl)triethoxysilane (APTES) treatment. This sensory fiber can selectively sense diethyl chlorophosphate even in the presence of interferrants like dimethyl methylphosphonate, triethyl phosphate, and tributylphosphate with a good linear range of 0 to 1.8 mM [91]. Personal health care devices, including smart watches and fabrics, are becoming increasingly popular. Conjugated polymers have shown promising results for the past few decades in health care and motion detection as well. PVDF nanofibers immobilized with polydiacetylene (PDA) monomers and its influence with solutions conductivity and viscosity were investigated. The morphological and phase conformations were established. The conjugated backbone of PDA twisted under high temperature, consequently reducing the conjugation length and increasing the band gap leading it to exhibit the colorimetric response from blue to red [92].

Recently, breath sensing was made possible with ES fibrous membranes comprising of polyacrylonitrile (PAN) NF with PANI shells. The conductive PANI shell thickness varied with in situ polymerization taking place on ES PAN fibers. Longer period leads to the agglomerated PANI and it reduces the atmospheric moisture contact with the fibrous membranes. PANI formed on the fibrous networks on interaction with atmospheric water vapor; their porosity shrunk and PANI elevates the current flow by reducing the resistance. By monitoring the current response, it is evident that humidity content and breath monitoring can be achieved [93].

Excessive consumption of Fe^2+^ can cause siderosis, damage the internal organs might lead to death. 4-vinylbenzylchloride polymerized through free radical polymerization to form poly(vinylbenzyl chloride) (PVBC) and it is electrospun into PVBC NFs. The PVBC NFs immobilized into 2-(2′-pyridyl)imidazole (PIMH) ligand solution acting as post functionalization treatment. The Fe^2+^ ion on interaction with PIMH ligand undergoes spin crossover from high spin Fe^2+^ to low spin Fe^2+^ recognized by characteristic absorbance peak. A distinct colorimetric response from colorless to reddish orange due to the formation of low spin six-coordinate complex without any considerable interference from other common metal ions (with negligible response to Ni^2+^) [86].

Many optical fiber-based solid-state sensors failed in stability especially in dye leaching test. Dye leaching is one of the major shortcomings in industrializing the optical sensors. Steyaert et al. overcome this issue by immobilizing the dye-functionalized copolymer into the ES polymeric carrier. A copolymer made of 2-hydroxyethyl acrylate (HEA) and commercial disperse red 1 modified as acrylate (DR1 A) using free radical polymerization. The leaching test performed with reference fabrics shows that copolymer dyes can outperform the other sensory fibers due to the very less stained observations. The developed sensory membrane shows good stability even under the higher pH 12 and its improved water fastness stability was characterized with hydrolysis % [94]. These interesting results shows the viability of producing the smart responsive garments can sense the presence of toxicants with the optical responses depending on the working environments [39].

Surface grafting helps to improve functionality, stability, and resistance to photobleaching and analyte diffusion. Surface grafting can be done on films or ES nanofibers. The covalent linking between the chromophore and the carrier polymer matrix might sufficiently elevate the sensor robustness against various environmental, natural, and artificial human interferences. Nitroaromatic compounds tend to create many ecological and health problems. Hidden explosives and terrorist attacks jeopardize global safety. 3,6-Dibromocarbazole-based monomer was treated with divinyl benzene and polymerized to polymer P. It was grafted on to the gelatin, taking advantage of the reaction between the active ester and free primary amine to form the resultant gel-p film (Figure 3b). The gelatin matrix comprised various amino acids and generated a rapid response mechanism. Gelatin’s tryptophan competes with oxygen, thereby preventing photobleaching. Gelation in the gelatin was governed by the amino and hydroxyl group in its triple helices that control the molecular configuration forbidding the photobleaching process [95]. The as-designed gel-p sensors exhibited ten cycles of repeatability and the process was reversed by blast drying through constant air circulation.

Alkaline phosphatase (ALP) sensing is needed because its excess level in human body can lead to bone disease, cancers, and diabetes. Monitoring the ALP concentration will be helpful in biomedical research sectors. Surface amination done on the ES polyethylene terephthalate (PET) fibers to easily graft with aldehyde functionalized fluorescein at room temperature and it is phosphorylated to form PET-Flu-PO4. The AIE active compound tetraphenylethene (TPE) was electrostatically complexed to form PET-Flu-PO4/TPE membranes with a good LOD of 1.5 mU/mL and producing the visible color change from blue to green in response to ALP presence with the optimal surface amine density of ~30nmol/mg [96]. More recently, Mao et al. overcome the limitation of high fluorine content polymer electrospinning by altering the viscosity of the precursor solution. High fluorine content is desirable for promoting the oxygen diffusion speed in the sensor matrices thereby achieving the high speed responsive sensors. The copolymeric system was adopted to improve the viscosity of the precursor solution leading to the stable fiber formation. Platinum porphyrin grafted co polymeric ES microfibrous system initially formed by free radical polymerization and it is electrospun [97].

Microbes are widely distributed in water and food resources, and their infections, due to improper sanitation facilities and personal healthcare, remain major causes of human death. Electrospun optical sensory fiber affords easy sensing of *Escherichia coli* with a sensing range of 10^2^ to 10^5^ colony-forming units (CFU)/mL. Polystyrene-co-maleic anhydride (PSMA) copolymer was electrospun and then immobilized with tetrakis(4-hydroxytetraphenyl)ethane cyanuric chloride (TPEC; prepared through the McMurry reaction) with three types of spacer to prevent unfavorable interactions with the ES fibrous structures (Figure 3c). Because of the *E. coli* protein site and the mannose hydroxyl group demonstrate a specific interaction, mannose was grafted on the aforementioned nanofibrous structures that restricted the TPEC intramolecular motions causing the AIE process and providing optical responses. Amino group densities and spacer molecule contribution also affect the sensing characteristics of the as designed sensors [90].

## 9. Reversible Addition Fragmentation Chain Transfer (RAFT) and Atom Transfer Radical Polymerization (ATRP)-Based ES Sensory Nanofibers

On contrast to free radical polymerization, controlled radical polymerization is reliable in creating well-defined polymers with controlled molecular weights, polydispersity index, and polymeric structures (block, star, hyperbranched, and cross-linked) with various functionalities [97,98,99]. The most widely employed techniques are reversible addition fragmentation chain transfer (RAFT) and atom transfer radical polymerization (ATRP). In RAFT polymerization, RAFT agent and initiator selection is highly important which possibly can curtail the self-initiation, side reactions, reaction kinetics, and polymer molecular weight. RAFT polymerization also affords better tolerance to various functional groups, facile control and operational environment.

Improving the sensor stability towards various organic solvents might also extend the sensor applicability in various environments. The sensing factors should never degrade on reaction with the external solvent medium. Cho et al. made a step forward in generating highly stable nanofibrous sensors towards water and various organic solvents. Poly(2-hydroxyethyl methacrylate-co-N-methylolacrylamide) (poly(HEMA-co-NMA)) synthesized using free radical polymerization and it is electrospun into nanofibers where HEMA imparts hydrophilicity and NMA provides stability to the ES fibrous structures. The poly(HEMA-co-NMA) ES fibers subjected to immobilization with ATRP initiators and further grafted with modified rhodamine/pyrene probes to generate the Hg^2+^ and Cu^2+^ sensors (Figure 4a). The resultant rhodamine grafted colorimetric behaviors observed from colorless to reddish orange transition on increasing the Hg^2+^ concentration from 10^−7^ M to 10^−2^ M without any interference from other common metal ions. Whereas, the pyrene grafted solid state sensor reveal its importance by producing visible color change from colorless to blue with increased Cu^2+^ concentration (10^−6^ M to 10^−1^ M) [100].

Graphene oxide (GO) RAFT was functionalized using 2-[(butylsulfanyl)-carbonothioylsulfanyl] propanoic acid (RAFT agent) and grafted with N-isopropylacrylamide (NIPAM) further hybridized with block copolymer to form a block copolymer brush-GO hybrid (Figure 4b). This novel complex hybrid could interact with Alq_3_ due to the presence of an 8-hydroxy quinolone unit. The fluorescent donor Alq_3_ possesses higher energy than the TNP analyte; this variation can lead to the excited state electron transfer harvesting substantial fluorescent turn off behavior [101]. Diblock amphiphilic polymer comprising thermoresponsive monomer N-isopropylacrylamide (NIPAM) and 6-[4-(4-sodium carboxylatephenylazo)phenoxyl] hexyl methacrylate (M6AzCOONa) were prepared using a convenient RAFT technique with the RAFT agent 2-cyanoprop-2-yl dithiobenzoate (CPDB). The carboxylate group of the substituent side chains can be protonated under acidic conditions making the azobenzene blocks hydrophobic and causing closer aggregations with better fluorescent emission intensities. Under basic conditions, self-organization and hydrogen bond formation eases the photoisomerization of block polymers. Poly NIPAM (PNIPAM) exhibits lower critical solution temperature (LCST) characteristics and the thermal treatment above the block PNIPAM LCST, causing a coil-to-globule transition and presenting rotation barriers that ultimately enhance fluorescent emission [102].

More recently, Taya et al. designed FRET based sensory device based on (poly(methyl methacrylate (MMA))-co-7-(4-trifluoromethyl)coumarin-N-methacrylamide (TCMA)) consisted of coumarin derivative as donor and curcumin fluorophore as an acceptor. Poly (MMA-co-TCMA) was synthesized by RAFT copolymerization using 4-cyano-4 (dodecylsulfanylthiocarbonyl)sulfanylpentanoicacid (CDP) as RAFT agent and the free radicals generated by 4,4_-azobis(isobutyronitrile) (AIBN). An ideal FRET pair was constructed due to the high quantum yield of polymer donor and the greater spectral overlap b/w donor and acceptor. Time-resolved fluorescence lifetime measurement reveals the average lifetime decrement with trinitrotoluene (TNT) analyte addition ascribed to the electron transfer from the lowest unoccupied molecular orbitals (LUMO) of the CP to the analyte. TNT specificity was higher on comparison to dinitrotoluene (DNT) due its higher electron affinity [69].

ATRP is a useful and approachable technique to introduce fluorophore into polymeric backbone to ensure the device phase stability required for maximizing luminescent solar concentrator performances [97]. Gu et al. designed aggregation caused quenching (ACQ) initiators, 4-(5-(4-(dimethylamino)phenyl)-1-phenyl-4,5-dihydro-1H-pyrazol-3-yl) phenyl-2-bromo-2-methylpropanoate (abbreviated as TPP-A) and introduced the initiator into polymer chains (PS or PNIPAM) via ATRP. The emission differences occurring with charge transfer, polarity, initiators, and temperature were contrasted, and it portrays the fine-tuning the polymer chains and ATRP initiators charge transfer [103]. Synthesis of linear and star block copolymers consisted of different arm numbers and block lengths made possible with ATRP. The stimuli responsiveness and its fluorescence response towards pH, and temperature was recorded and mechanisms proposed to account for such optical responses [104].

A short response sensory ES fibrous membrane fabricated with multi-arm copolymer prepared using ATRP. The copolymeric system and the chromophore are poly(isobutyl methacrylate)-co-poly(trifluoroethyl methacrylate)s (PolyIBMA-co-PolyTFEM)s and platinum porphyrin-based phosphorescence probes respectively. ES fibers displayed higher sensitivity as compared to sprayed films and shorter response time was harvested using high fluorine content because of its higher oxygen permeability [105]. Azide terminated diblock copolymer poly(N-isopropylacrylamide)-block-poly(N-methylolacrylamide) (PNIPAAm-b-PNMA) using ATRP and it is subjected to click reaction with alkynyl-terminated polyfluorene (PF) forming triblock rod–coil–coil (PF-b-PNIPAAm-b-PNMA) copolymer. Stearyl acrylate cross-linker was replaced with N-methylolacrylamide to form water stable ES fibrous membranes without causing any degradation to its fibrous morphology. The triblock copolymers prepared presents lower entanglements due to its lower molecular weights hence high molecular weight blending aids in producing stable ES fibers. Small-angle X-ray scattering (SAXS) and transmission electron microscopy (TEM) reveals the lamellar structure existence along the fiber axis. PNIPAAm blocks swell and light absorbed by polyfluorene (PF) fluorescent block creating a stronger fluorescent at high temperatures. The temperature responsive PL measurements remain reliable with very clean reversibility up to six cycles [106].

Fluorescent metal ion-sensitive poly(1-pyrenemethylmethacrylate) was alkynyl ended using ATRP technique and it underwent a click reaction by azide terminated poly(N-isopropylacrylamide)-block-poly(N-methylolacrylamide) to form the resultant multifunctional triblock (PPy-b-PNIPAAm-b-PNMA). This triblock was electrospun into solid state sensor and it can provide fluorescent responses to temperature and Fe^3+^ ions. The solvent effects leads to self- assembled structures in the ES fibrous membranes and it is ascribed to the chloroform poor solvent to PNMA block and it is a good solvent for the remaining two blocks. Increasing the blocks of PNMA cross-linker restricts the chain mobility leading to the reduced PL responsive behavior. The optimized molecular weights and PNMA blocks led to the successful development of Fe^3+^ and thermo sensory device with better reversibility and excellent water stability [107].

## 10. Random Copolymer Based ES Sensory Nanofibers

Conjugated copolymers can be designed either as random or block copolymers. Random copolymers synthetic process remains simple and it is easy to control. The as-synthesized copolymers can be exploited in the field of multisensory applications. Each unit in the copolymers can substantially contribute to the specific site for sensing the targeted analyte.

More recently, Akram et al. [108] utilized methyl methacrylate (MMA), octafluoropentyl methacrylate (OCFPM), pentafluorophenyl acrylate (PFPA), and PS were copolymerized with tetraphenylphorphinato platinum (II) monomer (PtTPP-MA) in different combinations to form random copolymers through free radical polymerization. Fluorinated monomer OCFPM and PFPA known for its higher oxygen affinity, in addition lowered photodegradation and temperature insensitiveness adding reliability to the as-designed oxygen sensors. A series of polymers generated, namely, poly(MMA-co-PtTPPMA) (P1), poly(PFPA-co-OCFPM-co-PtTPPMA) (P2), poly(PS-co-PFPA-co-OCFPM-co-PtTPPMA) (P3), and poly(MMA-co-PFPA-co-OCFPM-co-PtTPPMA) (P4), were designed. Furthermore, the polymers were transformed into thin film and ES nanofibers and their sensing parameters were contrasted significantly. Among which P1 failed to produce better sensitivity due to its intrinsic oxygen impermeability. Nanofibrous P3 polymer exhibits promising sensing response (4.42 s) due to the elevated oxygen permeation granted by the highly electronegative fluorinated monomers and amorphous blocks created numerous void volumes.

Taking advantage of NO gas-responsive 1,2-diaminoanthraquinone (DAQ) reaction in aqueous medium forms DAQ triazole derivative (red to yellow color), Chen et al. synthesized random copolymers of Poly((N-isopropylacrylamide)*-co-*(N-hydroxymethylacrylamide)) (poly(NIPAAm*-co-*NMA)) with NO (g)-responsive 1,2-diaminoanthraquinone (DAQ). The cross-linked NFs shows good stability and better color transitions were achieved with NFs instead of drop casted thin films [109].

Zn^2+^ ions play a key role as a co-factor in many hydroxylation enzymes and several biological transformations accompanied by signaling. Integrating HPBO into PNIPAAm could considerably respond to Zn^2+^ with respect to different pH and temperature by showing varied optical responses. Chen et al. synthesized random copolymers of poly{2-{2-hydroxyl-4-[5-(acryloxy)hexyloxy]phenyl}benzoxazole}-co-(N-isopropylacrylamide)-co-(stearyl acrylate)} (poly(HPBO-co-NIPAAm-co-SA)) using free-radical polymerization with different feed ratios of monomer to generated five type of polymers (P1-P5). HPBO is a well-known material for sensing Zn^2+^ or pH because of its structural changes produced by excited state intramolecular proton transfer (ESIPT) governing the stokes shift to harvest good optical responses. The optimized P4 nanofibers can detect Zn^2+^ at 10^-8^ M along with emission maximum shift of 60 nm engendering the naked eye visible color change. The reason for such good sensitivity is their excellent hydrophilicity. The random copolymer units consisted of PNIPAAm so that their LCST influences with the corresponding dimensional structural stability afforded by stearyl acrylate (SA) units were clearly demonstrated [110].

9,9-Dihexylfluorene-2-bipyridine-7-(4-vinylphenyl) (FBPY) homopolymer and its random copolymer synthesized, which is comprised of poly{(N-isopropylacrylamide)-co-(stearyl acid)-co-[9,9-dihexylfluorene-2-bipyridine-7-(4-vinylphenyl)]} (poly NIPAAm-co-SA-co-FBPY). FBPY units acts as a fluorescent part and its optical responses were unique in presence of Zn^2+^ ions. FBPY not only consist bipyridine unit, but it also contains vinylphenyl group, which is a prerequisite for free radical polymerization to form a random copolymer with SA and NIPAAm. Increment in SA composition leads to improved fiber diameter as a result of longer alkyl chains. The red shift observation with metal ion presence is attributable to the complex formation with bipyridine and promoting the conjugation with neighboring fluorophore [111].

Commercial rhodamine base treated with an excess of N-(2-hydroxyethyl)ethylenediamine in methanol medium to generate active fluorescent probe so-called spirolactam rhodamine derivative (SRhBOH). Different wt% of fluorescent probe SRhBOH was blended into poly(2-hydroxyethyl methacrylate-co-N-methylolacrylamide-co-nitrobenzoxadiazolyl derivative) (poly(HEMA-co-NMA-co-NBD)) random copolymer with an estimated composition of 10.26:1:0.1 to form the ES sensory nanofibers. A clean pH response with sensory fibers obtained with reducing the pH from 7 to 2 producing the visible red color transition. The reproducibility was excellent and it is monitored successfully by altering the pH from 12 to 2 reversibly. 20 wt% SRhBOH blended copolymers outperformed other blends by showing larger emission shift (57 nm). FRET process happen efficiently b/w NBD donor and ring opened SRhBOH acceptor supported by the extensive overlap between the PL spectrum of copolymer and UV–visible absorbance of chromophore. Reversible characters and color transition monitored with with international commission on illumination (CIE) coordinates add values to facile detection of Fe^3+^ selectively [112].

1-Benzoyl-3-[2-(2-allyl-1,3-dioxo-2,3-dihydro-1Hbenzo[de]isoquinolin-6-ylamino)-ethyl]-thiourea (BNPTU) and Fe_3_O_4_ nanoparticles (NPs) blended onto the free radical polymerized random copolymer poly(N-isopropylacrylamide)-co-(N-methylolacrylamide)-co-(acrylic acid) (poly(NIPAAm-co-NMA-co-AA)) to form the novel magnetic fluorescent sensory NFs. Because of the Hg^2+^ induced thiourea unit transformation into an imadazoline moiety under the water medium, the fluorescent color change was noticed without common metal ion interferences. The use of magnetic NPs aids in easy removal of contaminants from the water resources and sensory filter microfluidics system employed to simulate the real time hazardous ions monitoring [113].

A novel fluorescent probe N-(2-(3,6′-bis(diethylamino)-3-oxospiro[isoindoline-1,9′-xanthen]-2-yl)ethyl)methacrylamide (RhBN2AM) was synthesized from rhodamine base and it was free-radical polymerized along with 2-hydroxyethyl methacrylate (HEMA) and N-methylolacrylamide (NMA) to form a random copolymer, poly(HEMA-co-NMA-co-RhBN2AM). The as designed sensory fiber with the optimized compositions tends to produce colorless to red transition in response to Hg^2+^ aqueous solution. PL response was improved further on decreasing the pH from 7 to 2 with favorable reversible sensing capability [114]. Subsequently, as a further advancement, the hydrophilic HEMA moiety of poly(HEMA-co-NMA-co-RhBN2AM) random copolymer was altered with amphiphilic NIPAAm moiety to impart thermoresponsive characteristics due to its room temperature LCST (Figure 5). The novel fluorophoric group RhBN2AM did not respond to pH and metal ion changes under basic and neutral conditions. The sensory fiber changed color on acidic media and changed selectively in the presence of Hg^2+^ because of ring-opened spirocyclic form [115].

1,8-Naphthalimide-based monomer (BNPTU) FRET donor was prepared from 4-Bromo-N-allyl-1,8-naphthalimide, whereas the RhBAM-based monomer FRET acceptor was prepared from commercial rhodamine base. The acceptors and donors were framed into ES setup and it is spun into random copolymer as (poly(methyl methacrylatete-co-1,8-naphthalimide derivatives-co-rhodamine derivative); poly(MMA-co-BNPTU-co-RhBAM)) nanofibers. BNPTU units not only served as FRET donor, but also selectively sense Hg^2+^ due to the extensive conversion of thiourea unit into an imidazoline group rhodamine derivative is very well explored in pH sensing and here it also function as FRET acceptor. The colorimetric response varied rationally according to the Hg^2+^ concentrations and pH conditions. The color tuning is very precise and it is demonstrated with good repeatability supported by confocal and CIE graphs. Microfluidic system along with conductivity setup clearly demonstrates the higher surface to volume ratio afforded by the as-designed sensory fibers [116].

## 11. Block Copolymer Based ES Sensory Nanofibers

Several CPs with different binding sites and functionalized conjugated polymers proved to be applied as metal, volatile organic compounds (VOCs), pH, and temperature sensors [117,118,119,120]. Block copolymers (BCPs) can be facilely prepared and it is viable to form self-assemblies. Such beneficial BCPs can create highly ordered and tuned shape NPs acting as a templates. BCPs also finds interesting paths in creating highly efficient photovoltaics by forming self-organized microphase separations into a variety of desired nanostructures such as nanospheres, nanorods, nanofibrils, lamellar and micellar structures [121]. Even structural colors can be achieved by harnessing the regular periodicities and dielectric constants to produce mechanical force responses [122].

Tracing back to the evolution of sensory devices, sensing made simple by coupling the probe with receptor and the receptor site binds specifically to the desired analyte causing some electrical or optical responses. Optical responses were easily monitored without any expensive several complex circuits. Kuo et al. fabricated pH sensing luminescent nanofibers employing poly(phenylquinoline)-block-polystyrene (PPQ-b-PS). The effect of solvent and the ES fiber formation contributed significantly in reducing the PPQ aggregated size domains ultimately contributed to the efficient protonation thereby promoting the pH sensitivity [123].

Smart polymers can respond to the different stimuli imposed on it and, depending on the stimulant, the sensory architectures and monitoring devices varies. Material science and polymeric technology during several decades raised several methodologies to design such stimuli responsive polymers. These smart polymers can respond to a variety of external stimuli, such as optical, solvent exchange, electrical, thermal, mechanical stress, ion factor, redox, pH, chemical, environmental, and biological signals [27,107]. (PNIPAAm-b-PNMA) block copolymer synthesized using ATRP followed by click reaction with alkynyl-terminated polyfluorene (PF) generated a triblock rod–coil–coil (PF-b-PNIPAAm-b-PNMA) structure which is electrospun to produce the thermoresponsive fibers (Figure 6a). The thermoresponsive fluorescent behavior is clearly demonstrated due to the predominant LCST behavior supported by shrinking and swelling of the polymeric chains causing distinct PL responsiveness [106].

Very recently, Song et al. designed selective sensor for TNT detection with a temperature responsive block copolymer. RAFT initiated the synthetic process to prepare pyrene terminated block copolymer by utilizing the pyrene functional RAFT agent and NIPAM oligo(ethylene glycol) mono methyl ether methacrylate (OEGMA), 5-(2-methacryloyl-ethyloxymethyl)-8-quinolinol (MQ) monomers (functions as temperature probe). ZnS NPs amine groups possibly coordinate with MQ generating the fluorescent component. As a result, block copolymer decorated with ZnS NPs designated to cause fluorescent sensing responses on adding TNT because of FRET process. GO grafted onto the BCPs can also additionally improve the thermal stability. The fluorescence quenching succeeded because of FRET efficiency changes from fluorescent metal-quinolate complex centers to GO on PNIPAM establishing the thermosensitive polymeric shrinkage beyond LCST [27].

Block copolymers known to receive several attentions in sensing because of excellent avenues in developing multifunctional sensory systems. Following this trend, Wang et al. designed a block copolymer poly(1-pyrenemethyl methacrylate)-b-PNIPAAm-b-PNMA (PPy-b-PNIPAAm-b-PNMA) and its nanofibers found to exhibit multi-sensory applications (Figure 6b). The thermally cross-linked nanofibers small angle X-ray scattering (SAXS) results in first order peak with 19 nm d-spacing strengthening the formation of randomly oriented spherical nanostructures. The solvent compatibility plays a crucial role in achieving such unique interior nanostructured self-assembly within the fibers. The degree of higher cross-linking literally reduces the polymeric chain mobility hence swelling/ shrinking process of PNIPAAm affected the response characteristics. ES nanofiber displays highly linear blue fluorescence turn off response on comparison with thin film sensors [107].

Ren et al. utilized RAFT polymerization to exploit the blocks 6-[4-(4-sodium carboxylatephenylazo)phenoxyl] hexyl methacrylate (M6AzCOONa), which is pH- and UV-responsive and fluorescent, whereas NIPAM to impart thermoresponsive characters.

Temperature induced transition of coil to globule structure restricting the azobenzene’s molecular rotation to harvest turn off-on fluorescence. The as-synthesized blocks forms hydrogen bonding and self-organized structures under acidic and neutral medium which significantly forbids the cis-tans photoisomerization process whereas the chromophoric azobenzene remains free in basic conditions engenders the photoisomerization [102].

GO treated with RAFT agent 2-[(butylsulfanyl)-carbonothioylsulfanyl] propanoic acid forming GO-RAFT followed by PNIPAM grafting. The PNIPAM graft GO on free radical polymerization with glycidyl methacrylate (GMA) and 5-(2-methacryloyl-ethyloxymethyl)-8-quinolinol (MQ) can lead to the formation of PNIPAM-b-P(GMA-co-MQ) grafted GO hybrid and it is transformed into tris(8-hydroxyquinoline) aluminum (Alq_3_) containing block copolymer brush-GO hybrid. The sharp turn on–off responses obtained due to the FRET from the Alq3 emissive centers to GO possibly contributed by the PNIPAM conformational changes with respect to temperature. For the TNP detection, the frontier molecular orbitals contributed to the plausible mechanism for such selective optical behavior. The excited state electron transfer occurs from Alq_3_ to TNP analyte predominantly due to the higher LUMO level of fluorescent donor Alq_3_ to TNP. Other common electron deficient interfering analytes have considerably higher LUMO energy levels causing minor quenching alone [101]. The vast number of research works demonstrated the differences between the sensing ability of thin films and nanofibers [107,109]. ES nanofiber conquers the research community with its greater surface to volume ratio thereby providing good diffusion facilitating the exposure of plenty of active sites and interacts with the sensing analyte [105,108]. Finally, through consistent attempts researchers made good progress in confining the nanofiber dimensions that includes defect free diameter tunable with unique morphologies. As a consequence of ES evolution, the present status with sensitivity and response time factors were crucially reaching better designation. The chromophoric dye leaching, water and organic solvents stability issues with ES sensory NFs was noticed and addressed to a certain extent. The futuristic research will take account of several factors such as stability, reversibility, green material, cost factor, scalability, and safety concerns to greatly influence the ES sensory nanofibers commercialization in serving the mankind and the environment.

## 12. Futuristic Views on the Optical Sensory Fibers

As is evidenced from the discussed optical sensory fibers, environmental and health monitoring is an increasingly critical concern. Smart polymers can be efficiently converted to fibers that can be employed as smart garments [39,124]. Energy harvesters along with nanomolar and sub-nanomolar sensory device characteristics may take center-stage in the future. The CCP-based photovoltaics showed promising results in achieving higher fill factors. Such CCP ES-based light-emitting diodes might be the other plausible route to achieve success. Its exploration can also be extended to the popular topic emissive perovskite molecules. The reversible characteristic influences the sensory fibers in real-time applicability, and it is highly demanding to reduce the cost and material loss. The integration of thermoplastic copolymers into breathable wound dressing applications may have greater importance in textile-based studies [40,125,126]. Biodegradable and greener materials will most likely be used in creating sensory optical nanofibers in the near future. Stretchability, flexibility, robustness, and stability will become crucial concerns in integrating CCPs in various optoelectronic appliances.

## 13. Conclusions

Major breakthroughs observed in CCPs and their significance in terms of electrospun optical nanofibers were discussed with recent research outcomes. The promising and straightforward optical ES nanofibrous sensors displayed ultrahigh sensitivity and high-speed response due to the active-site exposure achieved with higher surface-to-volume ratio and better diffusion. The mechanisms underlying fluorescent and colorimetric responses were cohesively and concisely framed. Various synthetic and fabrication processes were contrasted, and their significance in generating the highly stable ultrasensitive high-speed responsive optical sensors was demonstrated. Moreover, the contributions of porosity, solvent, molecular weight, hydrophilicity, and other bonding factors altered the optical sensors’ performance considerably. CCPs have a promising future for use as stretchable and flexible sensors, field effect transistors, memory devices, light-emitting diodes, and other human interactive devices.

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
