# Peer review of "Conjugated Copolymers through Electrospinning Synthetic Strategies and Their Versatile Applications in Sensing Environmental Toxicants, pH, Temperature, and Humidity"

_polymers, 2020, doi:10.3390/polym12030587_

Round 1
Reviewer 1 Report
The manuscript lurks in a good review about sensory applications of conjugated polymers. However, there are too much errors in English that seriously down-grade its value. It seems that parts from each authors were simply put on it without checking contents. It should be modified via a professional English editing service, and only then go to peer-review. With current version, it is hard for me to give feedbacks to improve details of the review.
Reviewer 2 Report
In the reviewed manuscript, entitled „Conjugated Copolymers Synthetic Strategies Based on Electrospinning Techniques and Their Versatile Applications in Sensing Environmental Toxicants, pH, Temperature, and Humidity” the Authors have undertaken an ambitious task of summarising the published works dedicated to sensors that utilise conjugated polymer fibres. Unfortunately, the topic of the work has been defined too widely, which resulted in a rather shallow investigation of the research issue, yielding a popular scientific manuscript and one that is more popular than scientific. The state of the art, referring to electrospinning of conjugated polymers has been presented in a very general manner in the manuscript, with many of its aspects being omitted, e.g. the Authors have not discussed how structural modifications affect the electrospinning process of a given material.
The proposed classification of sensors that is based on the method of synthesising the polymers, appears artificial when being used for electrospinning, because the synthesis of a conjugated polymer does not have a direct impact to electrospinning of the conjugated polymer itself and if it does have such an impact, the Authors have not provided evidence to support such a claim and, therefore, this classification scheme.
The section dedicated to the description of the sensing properties is chaotic and poorly ordered. No comparison of the sensing properties of electrospun materials with those of unmodifiied materials or materials deposited using other techniques (i.e. yielding non-fibrous materials) has been included in the manuscript. The Authors, in their descriptions of sensing properties have commonly used such general phrases as :, improving its sensitivity and selectivity„ or „Blue ES nanofiber displayed greater sensitivity than red ES because of the presence of higher beta-phase fractions “
A great review consists of a clearly defined topic, carefully collected key articles, a discussion of the main areas of scientific debate, a summary of unsolved questions and a critical appraisal of implications for the field. Unfortunately, all those are lacking in the reviewed manuscript.
Reviewer 3 Report
The aim of this review is to highlight the significance of conjugated copolymeric systems and its various types of sensory applications. In addition, sensing trends in monitoring environmental toxicants, pH, temperature, and humidity are discussed.
The results presented in this paper are interesting; however, the authors need to consider the comments listed below before re-submission of the paper.
Major issues:
1. The authors should comment the influence of the distance between needle and grounded plate on the morphology and nanofiber dimension.
2. The charge of the deposited fibers has influence of the morphology of the fibers but there are other parameters such as solvent which also contribute. The authors should make some comments related to the solvent solvent factor on morphology of electrospun fibers.
Round 2
Reviewer 1 Report
The manuscript has been substantially modified and is now acceptable.
Reviewer 2 Report
(1) Authors have not discussed how structural modifications affect the electrospinning process of a given material. (2) The synthesis of a conjugated polymer does not have a direct impact to electrospinning of the conjugated polymer itself and if it does have such an impact, the Authors have not provided evidence to support such a claim.
Ans: Thanks for the reviewer comments. We underwent a wide literature survey in supporting our claim. We add the sentences as below, “The tight coiled conformations aroused several attempts in way to solve the brittleness and mechanical instability issue which is common in pristine CPs. Using insulating polymeric host mechanical stability was improved considerably without compromising the sensing factors [76]. The synergistic role of polydiarylfluorenes side chain length and ES stretching effect deliberately facilitates the polydiarylfluorene β phase formation demonstrating the elevated sensitivity [77]. poly[(9,9‐bis(3′‐(N,N‐dimethylamino)propyl)‐2,7‐fluorene)‐alt‐2,7‐(9,9‐dioctyl‐fluorene)] containing tertiary amine group physically interacts with carboxylated rubbers manifests the fluorescent elastomeric fibers with good stability and mechanical strength [78]. Fluorene containing poly(aryl ether nitrile) synthesized using aromatic nucleophilic substitution polymerization exhibits good solubility because of the existing bulky pendant groups. Such structural modifications positively contributed to electrospinning process, producing better thermal stability and water repellency [79].” (page no.8)
Reviewer Ans.: The proposed classification scheme for sensors (dependent on the polymerization procedure) is unjustified. The Authors still have not provided any evidence for any general dependences of sensor properties on the method of preparing the sensing layer. It would be necessary to include in a review a section devoted in general to such differences for electrospun and non-electrospun sensing layers.
(3) No comparison of the sensing properties of electrospun materials with those of unmodifiied materials or materials deposited using other techniques (i.e. yielding non-fibrous materials) has been included in the manuscript.
Ans: Thanks for the reviewer comments. The electrospinning acts as a versatile technique in demonstrating higher sensitivity due to the higher surface to volume ratio ans such observation was proved in a number of published research works. We support the sentence by the following, “The vast number of research works demonstrated the differences between the sensing ability of thin films and nanofibers [107, 109]. ES nanofiber conquers the research community with its greater surface to volume ratio thereby providing good diffusion facilitating the exposure of plenty of active sites and interacts with the sensing analyte [105, 108].”(page no. 20)
Reviewer Ans.: In this case, the Authors have also not presented a comparison of classical materials with electrospun materials, e.g. in the form of a Table, which would contain information about the postulated (by each reference item) effect of the synthetic procedure (as was the proposed classification scheme) on the response of the sensor manufactured using ES, as opposed to those of sensors produced using classical methodology.
(4) The Authors, in their descriptions of sensing properties have commonly used such general phrases as :, improving its sensitivity and selectivity„ or „Blue ES nanofiber displayed greater sensitivity than red ES because of the presence of higher beta-phase fractions.
Ans: Yes we agree with the reviewer comment and we made correction as follows, Blue phase enriched PDA ES nanofiber displayed greater sensitivity than red phase PDA because of the presence of higher beta-phase fractions [75]. (page no.8)
Reviewer Ans.: The cited fragment was an example of an issue prevalent in the manuscript, given to illustrate to the Author the overly-general nature of some phrases. All similarly generalized phrases should be modified, not just this one example.
(5) The main areas of scientific debate, a summary of unsolved questions and a critical appraisal of implications for the field. Unfortunately, all those are lacking in the reviewed manuscript.
Ans: Thanks for the comments. We herein make an effort to furnish the simple and clear understanding of the ES based sensory NFs. We discussed the basics and the factors influencing it with the possible mechanism for such sensory responses. The major backlogs of selectivity, sensitivity, leaching, stability were considered and its improvements were compiled using different strategies of designing the sensory NFs. Classification of ES-based optical sensor fabrication and its potential in overcoming the existing sensory challenges such as sensitivity, response time, selectivity, dye leaching, instability, and reversibility are also discussed. “CPs finds difficulty in generating ESNFs due to its poor miscibility so that CPs were framed as CCPs by grafting, adding functionalities onto the monomers and preparing random or block copolymers to employ them as sensors.” (page no.4) “Finally, through consistent attempts researchers made good progress in confining the nanofiber dimensions that includes defect free diameter tunable with unique morphologies. As a consequence of ES evolution, the present status with sensitivity and response time factors were crucially reaching better designation. The chromophoric dye leaching, water and organic solvents stability issues with ES sensory NFs was noticed and addressed to a certain extent. The futuristic research will take account of several factors such as stability, reversibility, green material, cost factor, scalability and safety concerns to greatly influence the ES sensory nanofibers commercialization in serving the mankind and the environment.” (page no.20)
Reviewer Ans.: The presented summary does not constitute any basis for solving further scientific problems by researchers and cannot constitute a tutorial review for entry level students and researchers, as it is too shallow for an in-depth review article on detecting a particular analyte and too chaotic for a tutorial article.
In my opinion, the manuscript does not fulfill the abovementioned requirements posed to a good literature review. Consequently, I must advise against the publication of this manuscript in MDPI Polymers.
Reviewer 3 Report
The authors revised the manuscript as per reviewer's comments. The amended manuscript is acceptable for publication.